# Antioxidants and azd0156 Rescue Inflammatory Response in Autophagy-Impaired Macrophages

**DOI:** 10.3390/ijms25010169

**Published:** 2023-12-21

**Authors:** Abdalla Elbialy, Mai Kitauchi, Dai Yamanouchi

**Affiliations:** Division of Vascular Surgery, Department of Surgery, University of Wisconsin School of Medicine and Public Health, 1111 Highland Avenue, WIMR 5151, Madison, WI 53705, USA; aelbialy@wisc.edu (A.E.); me502031@s.okayama-u.ac.jp (M.K.)

**Keywords:** abdominal aortic aneurysm, macrophage, autophagy, ataxia telangiectasia mutated, NF-kB

## Abstract

Autophagy is a lysosomal degradation system that eliminates and recycles damaged intracellular organelles and proteins. Inflammatory macrophages play a critical role in the development of various age-related inflammatory illnesses such as abdominal aortic aneurysm, atherosclerosis, and rheumatoid arthritis; therefore, identifying the mechanisms that cause macrophage inflammation is crucial for a better understanding of and developing therapeutics for inflammatory diseases. Previous research has linked autophagy to macrophage inflammation; Atg16L1-deficient macrophages increase IL-1 and IL-18 production via inflammasome activation. In this study, however, we show an alternative pathway of macrophage inflammation in an autophagy-deficient environment. We found that inhibiting autophagy in THP1 macrophages progressively increased the expression of p65-mediated inflammatory genes. This effect was reversed by treatment with antioxidants or azd0156, an ataxia telangiectasia mutated (ATM) inhibitor. In addition, our results showed that M1 macrophages inhibit autophagy and induce DNA damage, whereas M2 macrophages activate autophagy and reduce DNA damage. Importantly, the chemical activation of autophagy or ATM inhibition during M1 polarization reduced the M1 phenotype and inflammation, whereas inhibiting autophagy during M2 polarization also reduced the M2 phenotype. Thus, our findings highlight the importance of the autophagy–ATM pathway in driving macrophage inflammation.

## 1. Introduction

Monocytes/macrophages have a remarkable degree of plasticity in tissues and can quickly adapt to a wide range of environmental cues [1]. In vitro traditional macrophage classification is oversimplified into the M1/M2 phenotypes, wherein M1-like, or classically activated, inflammatory macrophages secrete proinflammatory cytokines such as IL-1B, TNF, and INOS. M2-like macrophages, also known as alternatively activated macrophages, have anti-inflammatory properties and are involved in extracellular matrix remodeling [2]. Monocyte/macrophage inflammation is a cornerstone of many age-related inflammatory illnesses and autoimmune diseases such as abdominal aortic aneurysm (AAA), atherosclerosis, multiple sclerosis, and rheumatic arthritis [3,4,5,6,7], and identifying the pathways that cause macrophage inflammation is thus a vital step in understanding age-related inflammatory diseases and developing effective therapies.

On the other hand, autophagy is an essential mechanism for maintaining cellular homeostasis by recycling damaged proteins and organelles within the cell. Autophagy primarily targets protein aggregates, damaged mitochondria, unnecessary peroxisomes, excess ribosomes, the endoplasmic reticulum, endosomes, lipid droplets, and intracellular pathogens [8]. The accumulation of damaged proteins and organelles as a result of autophagy failure plays a role in the pathogenesis of many age-related disorders, including neurodegeneration, cardiovascular disease, cancer, inflammatory bowel disease, and others [3]. Autophagy is commonly diminished in both normal and pathological aging [7,8].

Tatsuya Saitoh et al. [9] reported a relationship between autophagy and inflammasomes in macrophages. They found that Atg16L1-defective macrophages increase IL-1 and IL-18 production but not LPS-induced IL-6 or TNF-α production. Further studies uncovered the underlying mechanisms that mitochondrial-derived DAMPs (damaged-associated molecular patterns) are the activators of inflammasomes [10,11].

In my previous publication, I studied the downstream pathway of autophagy and how autophagy deficiency contributes to harmful effects on fibroblasts [12]. The study used bioinformatics tools and fibroblast immunostaining to investigate the effects of autophagy suppression on cellular processes and signaling pathways. The study found that autophagy deficiency increased p38 signaling, DNA damage marker H2A.X, and oxidative stress marker dityrosine. The study found that antioxidants could reverse the harmful effects of autophagy suppression on p38 signaling and DNA damage response, indicating the association between oxidative stress and autophagy suppression.

In the current study, we will test whether inflammation is induced in the same way in autophagy-deficient macrophages and whether oxidative stress and/or DNA damage mediate macrophage inflammation and phenotypes.

## 2. Results

### 2.1. M1 Macrophages Suppress Autophagy and Autophagy Induction in M1 Macrophages Reduces Inflammation

We have recently published a new way for quantifying autophagy in vivo utilizing 3D image analysis and avoiding nucleus labeling after autophagy marker staining. Since 2D immunofluorescence images can only provide a qualitative analysis of protein expression, our method provided an accurate approach for quantification of p62/SQSTM1 puncta [12]. Macrophage phenotypes are categorized into M1 (inflammatory) and M2 (anti-inflammatory). To see if autophagy flux differs between the two, we assessed p62/SQSTM1 puncta in M1 and M2 macrophages. The autophagy marker p62/SQSTM1 binds to damaged proteins and can indicate the autophagic state. Autophagy activation is associated with lower p62 protein levels, and autophagy inhibition is associated with higher P62 levels, thus cytoplasmic p62 puncta are used to detect the autophagy level [12]. To evaluate autophagic carrier flux, p62 puncta were evaluated in the presence of bafilomycin (2 h), a lysosomal degradation inhibitor (Figure 1A). Importantly, M1 macrophages significantly increased p62 puncta, indicating a robust suppression of autophagy (Figure 1A). Additionally, *SQSTM1* gene expression was significantly upregulated in M1 macrophages (Figure 1B).

To investigate whether autophagy suppression in M1 is linked to the inflammatory phenotype, we activated autophagy using hemin during M1 transition. Hemin is a well-known inducer of autophagy via the LRP1 receptor [13]. Intriguingly, activating autophagy during the M1 transition reduced the expression of M1 macrophage marker *INOS* and increased the expression of M2 macrophage marker *ARG1* (Figure 1C). Furthermore, activating autophagy during the M1 transition reduced the expression of NF-kB p65 inflammatory genes (Figure 1D).

### 2.2. Autophagy Dysfunction Drives THP1 Macrophage Cell Inflammation in an Oxidative-Stress-Dependent Manner

Since autophagy activation reduces macrophage inflammation, we investigated if autophagy suppression drives inflammation in THP1 macrophage cells. Bafilomycin (Baf) is commonly used 2 h before analysis to assess autophagy flux. Prolonged incubation with Baf, on the other hand, is considered a strong autophagy inhibitor since it completely blocks the autophagy pathway by preventing autophagosomal breakdown [14]. To investigate the effect of autophagy suppression on macrophage NF-kB p65 inflammatory genes, we first treated undifferentiated THP1 macrophage cells for 72 h with Baf (100 nM). Unexpectedly, Baf treatment increased the expression of NF-kB p65 inflammatory genes by hundreds of times (Figure 2). In a recent article [12], we found that the majority of the deleterious effects of autophagy suppression in zebrafish fibroblasts, such as DNA damage and p38 signal activation, are mediated by oxidative stress induction [12]. To test whether macrophage-derived inflammation in Baf-treated THP1 cells is mediated by oxidative stress, we added an antioxidant N-acetyl cysteine (NAC) to Baf-treated cells. All NF-kB p65 inflammatory genes were restored to normal following NAC treatment, demonstrating that oxidative stress promotes inflammation in autophagy-deficient macrophages (Figure 2). However, a 24 h incubation with 3-Methyladenine (3MA) is frequently utilized to suppress autophagy. 3MA inhibits the early stages of autophagy by inhibiting class III PI3 [14]. We replaced 72 h of Baf treatment with 24 h of 3MA (2.5 mM) to see if we could achieve the same findings with a different autophagy inhibitor (Figure 3). 3MA increased NF-kB p65 inflammatory gene expression (Figure 3A) as well as NF-kB p65 protein expression (Figure 3B). The expression of the NF-kB p65 protein and its inflammatory genes were restored by NAC treatment, suggesting that oxidative stress promotes inflammation in autophagy-deficient macrophages (Figure 3).

### 2.3. Autophagy Deficiency and M1 Macrophage Cells Induce DNA Damage in an Oxidative-Stress-Dependent Manner

In our previous publication, we showed that autophagy-deficient fibroblasts increased DNA damage and the DNA damage sensor ataxia telangiectasia mutated (ATM) in an oxidative-stress-dependent manner [12]. Since autophagy-deficient THP1 cells increased oxidative stress, we investigated whether oxidative stress mediates inflammation in THP1 macrophages via the ATM pathway. First, we used a DNA damage marker (H2A.X) to detect DNA damage in THP1 cells upon 3MA treatment. As shown in Figure 4A, H2A.X-stained cells were increased by 3MA treatment and were restored to normal levels by using NAC antioxidant, indicating that oxidative stress promotes DNA damage in autophagy-defective macrophages. Furthermore, M1 macrophages had more H2A.X-labeled cells than M2 macrophages (Figure 4B). Intriguingly, activating autophagy with hemin or using antioxidant NAC during M1 polarization decreased H2A-labeled cells, whereas suppressing autophagy during M2 polarization increased H2A-labeled cells (Figure 4B). Similarly, quantitative PCR (QPCR) analysis showed that M1 macrophages induced the expression of DNA damage markers more significantly than M2 macrophages. Additionally, activating autophagy with hemin during M1 polarization decreased the expression of DNA damage marker genes, whereas suppressing autophagy during M2 polarization with bafilomycin induced the expression of DNA damage marker genes (Figure 4C).

### 2.4. ATM Inhibitors Alleviate Inflammation in Autophagy-Deficient and M1 THP1 Macrophage Cells

Next, we investigated whether the DNA damage sensor ATM is involved in mediating the expression of inflammatory genes in autophagy-deficient and M1 macrophages. ATM inhibition using azd0156 suppressed NF-kB p65 inflammatory genes (Figure 3A) and NF-kB p65 protein expression in M1 THP1 macrophage cells (Figure 3B) in 3MA-exposed macrophages. Here, we investigated the effect of ATM inhibition in THP1 macrophage cells after M1 polarization. M1-differentiated THP1 macrophage cells were treated with ATM inhibitor azd0156. ATM inhibition using azd0156 significantly suppressed NF-kB p65 inflammatory genes to a normal level in M1 inflammatory macrophages (Figure 5).

### 2.5. Autophagy–ATM Pathway Regulates AAA Development In Vivo

Since inflammatory macrophages plays an important role in AAA development and the autophagy–ATM pathway regulates macrophage inflammatory gene expression, we investigated whether the autophagy–ATM pathway regulates AAA development. We induced an AAA mouse model by exposing the aorta to CaCl2. To inhibit autophagy and ATM in vivo, mice were administered 3MA and azd0156 during surgery [15]. One week after surgery, the diameter of the aorta was measured in millimeters (mm), and the arteries were collected for histology and qPCR analysis (Figure 6).

Interestingly, the inhibition of ATM using adz0156 completely prevented the development of AAA and restored the diameter of the artery to normal (Figure 6A,B). The administration of 3MA in the artery increased aorta size (Figure 6A,B) and leukocytic and macrophage infiltration, as shown by *Cd45* and Cd68, respectively, as well as the expression of matrix metalloproteinases (*Mmp2*, *Mmp9*), inflammatory genes (*Cxcl2*, *Ccl4*, *Thbs1*), and DNA-damage-related genes (*Ung*, *Ddb1*) (Figure 6C–E). Importantly, the inhibition of ATM in 3MA-exposed mice maintained tunica media integrity, reduced the size of the tunica adventitia, as shown in Figure 6A, and restored aorta size (Figure 6A,B) and the expression of matrix metalloproteinases (Mmp2, Mmp9), inflammatory genes (*Cxcl2*, *Ccl4*, *Thbs1*), and DNA-damage-related genes (*Ung*, *Ddb1*), as well as reduced macrophage infiltration (Figure 6C–E). These findings show that ATM is essential for AAA development and that inhibiting ATM in the presence or absence of an autophagy inhibitor restored AAA development and that the autophagy–ATM pathway regulates AAA development.

These results suggest that ATM inhibitors suppress inflammation in autophagy-deficient and M1 THP1 macrophage cells in an oxidative-stress-dependent manner.

## 3. Discussion

Macrophage activation play a crucial role in inflammatory responses and is instrumental in the development of inflammation and associated diseases, including AAA [16,17]. The presence of activated macrophages in AAA contributes to the progression and exacerbation of aortic tissue degeneration, highlighting the importance of understanding the underlying mechanisms and potential therapeutic targets for the management of AAA [4,5,6,7]. They are an essential component of the immune system and contribute significantly to the pathogenesis of inflammatory and autoimmune disorders. Therefore, identifying the pathways that cause macrophage inflammation is a vital step in understanding inflammatory, autoimmune diseases and developing effective therapies.

Autophagy is a highly conserved cellular process responsible for the removal and recycling of damaged cellular organelles, thus maintaining cellular homeostasis [18]. In our previous publications, we have established an accurate method for in vivo autophagy quantification through a 3D analysis of immunostaining images stained with a specific autophagy marker [12]. Previous research has found a link between reduced autophagy and macrophage inflammation [9,11,19]; it was revealed that Atg16L1-defective macrophages increase IL-1 and IL-18 production via the DAMPs–inflammasome pathway [9,10,11]. In a previous publication, we showed that autophagy-deficient fibroblasts induced ATM and DNA damage in an oxidative-stress-dependent manner [12].

In this study, we investigated whether our previously identified mechanism in the autophagy-deficient environment is linked to macrophage inflammation. We explored the role of M1 macrophages in autophagy suppression and the subsequent effects on inflammation and DNA damage, with the aim of identifying potential therapeutic targets for AAA [20]. We employed a novel 3D image analysis technique to quantify autophagy in vivo [12], which allowed for a more accurate assessment of p62/SQSTM1 puncta in M1 and M2 macrophages, revealing that M1 macrophages significantly increased p62 puncta, indicating a robust suppression of autophagy. Additionally, *SQSTM1* gene expression was significantly upregulated in M1 macrophages [8].

We further explored whether autophagy manipulation in M1 macrophages was linked to the inflammatory phenotype [4,5,6,7]. Autophagy activation using hemin during M1 transition resulted in the reduced expression of M1 macrophage marker *INOS* and increased expression of M2 macrophage marker *ARG1* [13]. This activation also led to the decreased expression of NF-kB p65 inflammatory genes [20]. Conversely, M2 macrophages were found to preserve autophagy and decrease DNA damage, while inhibiting autophagy during M2 polarization reduced the M2 phenotype and increased inflammation.

To understand if autophagy suppression drives inflammation in THP1 macrophage cells, we treated undifferentiated THP1 macrophage cells with bafilomycin and observed an increase in the expression of NF-kB p65 inflammatory genes [20]. This effect was reversed by adding the antioxidant NAC [12], demonstrating that oxidative stress promotes inflammation in autophagy-deficient macrophages. Similar results were obtained using a different autophagy inhibitor, 3MA. We also found that autophagy deficiency and differentiation into M1 macrophages led to DNA damage in an oxidative-stress-dependent manner, as demonstrated by the increase in H2A.X-labeled cells [12]. Activating autophagy with hemin or using antioxidant NAC during M1 polarization decreased H2A-labeled cells, while suppressing autophagy during M2 polarization increased H2A-labeled cells [14].

Finally, we investigated the involvement of the DNA damage sensor ATM in mediating the expression of inflammatory genes in autophagy-deficient and M1 macrophages [12]. ATM inhibition using azd0156 suppressed NF-kB p65 inflammatory genes and protein expression in 3MA-exposed macrophages and M1-differentiated THP1 macrophage cells. These results suggest that ATM inhibitors suppress inflammation in autophagy-deficient and M1 THP1 macrophage cells.

Because our findings were based on in vitro THP1 macrophage experiments, it is critical to test this pathway in macrophages in vivo mouse AAA models, as well as in inflammatory diseases associated with macrophage inflammation. It is well established that autophagy levels generally decline with both normal and pathological aging [12]. It would indeed be intriguing to investigate whether the relationship between aging and AAA can be explained by autophagy deficiency. Consequently, it is imperative to investigate the role of the autophagy–ATM pathway in the context of aging and age-related inflammatory diseases in vivo. Understanding the interplay between autophagy, ATM signaling, and inflammation in aging could pave the way for a better comprehension of the relationship between AAA and aging. Investigating the complex interactions among these cellular processes may uncover key mechanisms that contribute to AAA development and progression in older populations. Further studies exploring these potential connections can offer valuable insights into the pathophysiology of AAA and help identify potential therapeutic targets for the prevention and treatment of this condition in aging populations.

## 4. Materials and Methods

### 4.1. Human THP1 Cell Culture and Polarization

THP1 macrophage cells (TIB-202, ATCC, Manassas, VA, USA) were cultured in RPMI-1640 (11875093, Thermofisher Scientific, Waltham, MA, USA) medium supplemented with 10% (*v*/*v*) not heat-inactivated fetal bovine serum (FBS) (30-2020, ATCC), 1% (*v*/*v*) penicillin-streptomycin (15140122, Gibco, Waltham, MA, USA), and 0.05 mM 2-mercaptoethanol (21985023, Thermofisher Scientific) at 37 °C, sustained with a 5% CO_2_ supply.

THP1 cells were treated with phorbol 12-myristate 13-acetate (PMA) (1652981, Biogems, Westlake Village, CA, USA) (50 ng/mL) for 48 h and then cultured in PMA-free medium for 24 h to obtain undifferentiated macrophages (M0). M1 macrophages were obtained by adding interferon-gamma (IFN-g) (25 ng/mL) (300-02, PeproTech, Cranbury, NJ, USA) and lipopolysaccharide (LPS) (100 ng/mL) (L6529, Sigma-Aldrich, St. Louis, MO, USA) to M0 for an additional 24 h, whereas M2 macrophages were obtained by adding interleukin-4 (IL-4) (20 ng/mL) (200-04, PeproTech) and interleukin-13 (IL-13) (20 ng/mL) (AF-200-13, PeproTech) for 24 h.

### 4.2. Chemicals

To evaluate autophagic flux, THP1 cells were treated with bafilomycin (sc-201550, Santa Cruz, Santa Cruz, CA, USA) (100 nM) for 2 h. For the inhibition of oxidative stress, THP1 cells were treated with NAC (25 mM) (A7250, Sigma-Aldrich) for 72 h. For autophagy inhibition, M2 and undifferentiated THP1 cells were treated with bafilomycin (100 nM) for 72 h or 3MA (2.5 mM) (M9281, Sigma-Aldrich). For autophagy activation, M1 THP1 cells were treated with hemin (25 μM) (4741, Sigma-Aldrich) for 24 h. For ATM inhibition, M1 THP1 cells were treated with azd0156 (10 nM) (4741, Selleckchem, Houston, UT, USA) for 24 h.

### 4.3. CaCl2-Induced AAA Mouse Model

The CaCl2-induced AAA mouse model has been generated as previously described [21]. 

Male, 12-week-old C57BL/6 mice were purchased from Jackson Laboratory (Bar Harbor, ME, USA). All mice had free access to a normal diet and water. The mice were injected intraperitoneally (IP) with an ATM inhibitor (azd0156) and/or an autophagy inhibitor (3MA), 200 µL each per mouse. 

Under general anesthesia, the infrarenal region of the abdominal aorta was isolated through a midline incision. A small piece of gauze soaked in 0.5 mol/L CaCl2 was applied perivascularly for 10 min. This gauze was then replaced with another piece of PBS-soaked gauze for 5 min. One week after surgery, the diameter of the aorta was measured in millimeters (mm), and the arteries were collected for histology and qPCR analysis.

### 4.4. Quantitative PCR (qPCR)

THP1 cells were cultured in 6-well plates 24 h before the treatment. The various chemicals were then incubated as previously stated. QPCR was carried out in triplicate. Total RNA was isolated from undifferentiated M0, M1, and M2 THP1 cells using Trizol reagents (15596026, Invitrogen, Waltham, MA, USA) according to the manufacturer’s instructions, with one exception: instead of precipitating isolated RNA from the supernatant with isopropanol, the supernatant was transferred to spin columns of pure link RNA kits (12183018A, Thermofisher Scientific), and RNA was collected as instructed. To synthesize cDNA, the All-In-One 5X RT MasterMix (G592, Abm, Burnaby, BC, Canada) was used. All reactions were carried out in a 20 L volume with 10 µL of SYBR^®^ Green JumpStart™ Taq ReadyMix™ (S4438, Sigma-Aldrich), 1 µL of each primer (10 µM), 1 µL cDNA, and 0.2 µL ROX Reference Dye. The qPCR primer sequences used in this study were retrieved from Origene. The housekeeping genes glyceraldehyde-3-phosphate dehydrogenase (*GADPH*) and actin beta (*ACTB*) were used as internal controls. Thermal cycling was carried out under the following conditions: initial denaturation—95 °C for 5 min; denaturation—95 °C for 15 s; annealing—60 °C for 1 min; melting curve—95 °C for 15 s, then 60 °C for 15 s, then 95 °C for 15 s. QPCR was performed using the 7500 fast real-time pcr system (Applied Biosystems, Foster City, CA, USA). We retrieved qPCR primer sequences used in this study from ORIGENE (https://www.origene.com/ accessed on 20 January 2023), and the human and mouse primer sequences are listed in Appendix A, respectively.

### 4.5. Immunocytochemistry

THP1 cells were grown in 6-well plates for 24 h prior to treatment. The various chemicals were then incubated as previously stated. THP1 cells were washed once with PBS before being fixed for 15 min at room temperature in PBS/4% paraformaldehyde (PFA). Cells were then washed and collected in PBS/40% fetal bovine serum (FBS) (35-011-CV, Corning, New York, NY, USA). After that, cells were plated as small drops onto microscope slides (12-550-15, Fisher Scientific) and air dried. Distilled-water-washed cells were then permeabilized at room temperature in PBS/0.2% Triton X-100 (X100, Sigma-Aldrich) for 20 min. Slides were then washed three times in PBS for 5 min then blocked in PBS/0.5% bovine serum albumin (BSA) (A7906, Sigma-Aldrich) for 1 h. Slides were then incubated with anti-RELA/NFκB p65 antibody (F-6) (sc-8008, Santa Cruz), anti-p62/SQSTM1 antibody (NBP148320, Novus Biologicals, Centennial, CO, USA), or anti-Phospho-Histone H2A.X (Ser139) antibody (14-9865-82, Invitrogen) in blocking buffer at a 1/100 dilution overnight at 4 °C. Slides were washed three times in PBS for 5 min then incubated with secondary antibodies m-IgG Fc BP-CFL 488 (sc-533653, Santa Cruz) or Donkey anti-Rabbit IgG (H + L) Highly Cross-Adsorbed Secondary Antibody, Alexa Fluor™ 594 (A-21207, Thermo Scientific) in blocking buffer at 1/100 dilution in the dark at room temperature for 2 h. Slides were then washed three times in PBS for 5 min and counterstained in the dark with DAPI (D1306, Thermo Scientific) at a dilution of 1/250 for 5 min before being mounted with Fluoromount (Sigma-Aldrich, F4680).

## 5. Conclusions

In conclusion, our study investigated the role of autophagy in M1 macrophages, inflammation, and DNA damage, with the aim of identifying potential therapeutic targets for AAA. We discovered that autophagy is suppressed in M1 macrophages, contributing to inflammation and DNA damage. Furthermore, we demonstrated the involvement of DNA damage sensor ATM signaling in mediating inflammatory gene expression. Both autophagy activation and ATM inhibition led to the suppression of inflammation and DNA damage. Future research should validate these findings in in vivo models and explore the relationship between aging, autophagy deficiency, and AAA development.

## Figures and Tables

**Figure 1 ijms-25-00169-f001:**
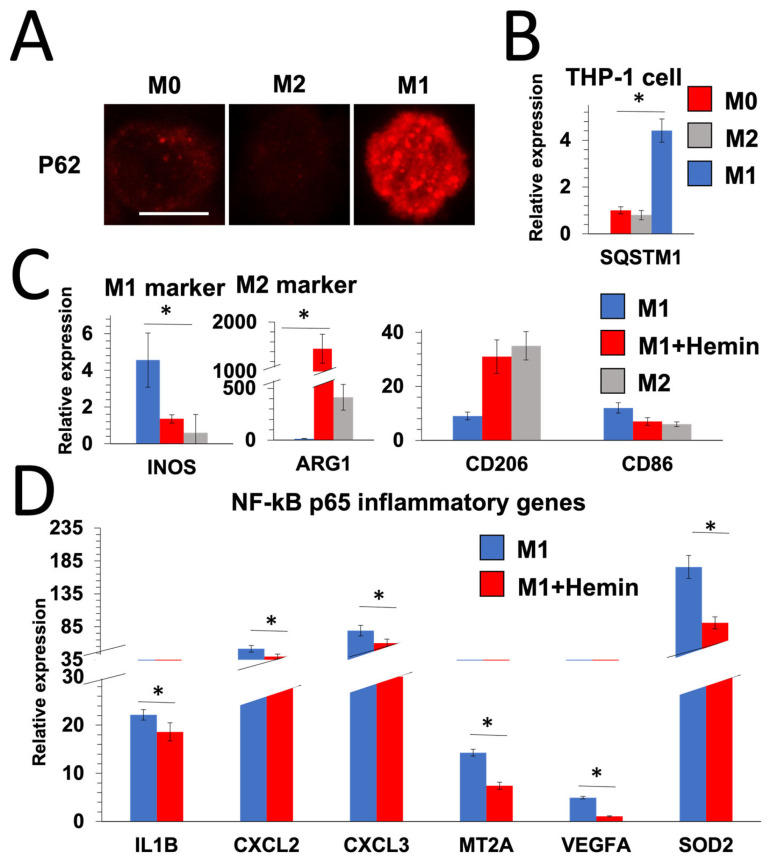
Autophagy induction in M1 macrophages reduce inflammation. Bar = 10 μm. (**A**) Representative 3D images of p62 immunostaining in M0, M1, and M2 macrophages. p62 was evaluated in the presence of bafilomycin, a lysosomal degradation inhibitor. (**B**–**D**) Levels of expression of *SQSTM1*, M1, and M2 macrophage markers and NF-kB p65 inflammatory genes in M0, M1, and M2 THP1 cells following hemin treatment (25 µM) (autophagy activator) (*n* = 3) (* *p*-value < 0.05, one-way ANOVA).

**Figure 2 ijms-25-00169-f002:**
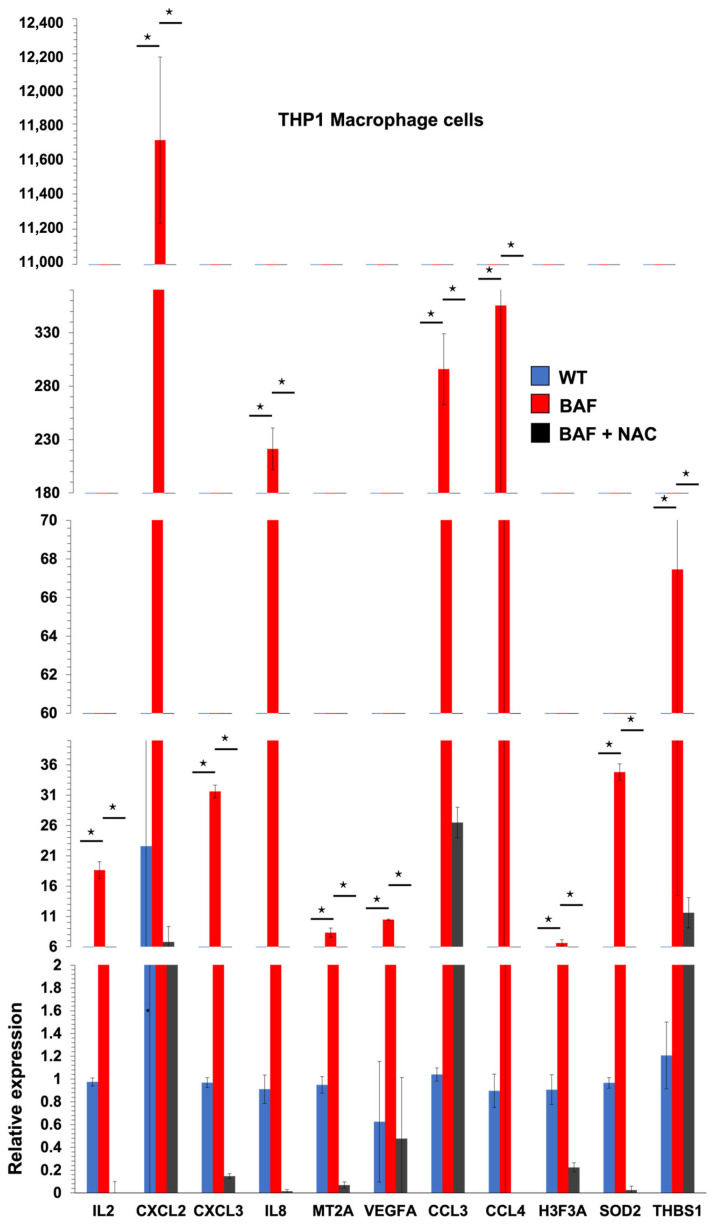
Autophagy dysfunction drives THP1 macrophage cell inflammation in an oxidative-stress-dependent manner: Levels of expression of NF-kB p65 inflammatory genes in undifferentiated THP1 cells (*n* = 3) after treatment with bafilomycin (100 nM) and NAC (25 mM) for 72 h. All genes were significantly upregulated upon bafilomycin treatment (* *p*-value < 0.05, one-way ANOVA).

**Figure 3 ijms-25-00169-f003:**
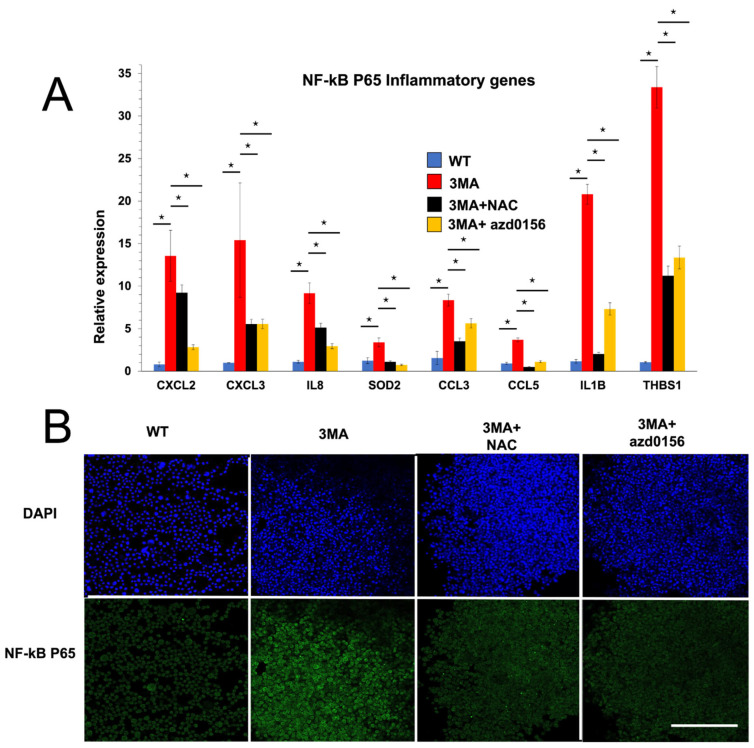
Autophagy dysfunction drives THP1 macrophage cell inflammation in an oxidative-stress—ATM-dependent manner: (**A**) Levels of expression of NF-kB p65 inflammatory genes in undifferentiated THP1 cells (*n* = 3) after 24 h of treatment with 3MA (2.5 mM), NAC (25 mM), and ATM inhibitor azd0156. All genes were significantly upregulated upon 3MA treatment (* *p*-value < 0.05, ANOVA). (**B**) NF-kB p65 immunostaining of undifferentiated THP1 cells after 24 h of treatment with 3MA, NAC, and ATM inhibitor azd0156. Bar = 200 µm.

**Figure 4 ijms-25-00169-f004:**
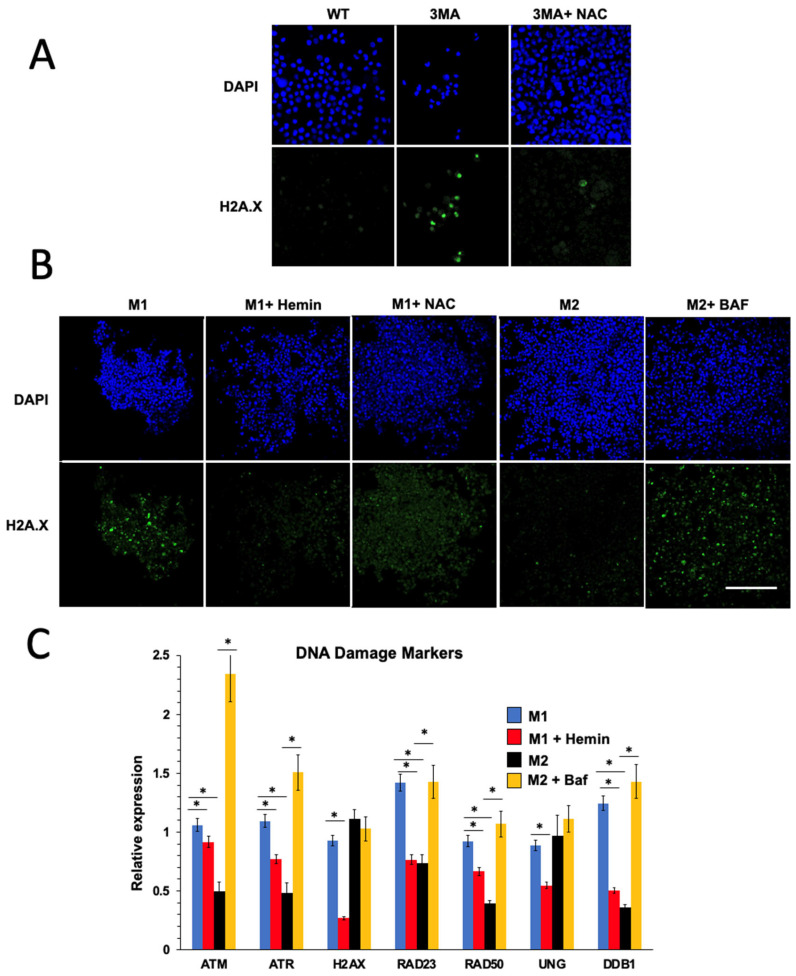
Autophagy deficiency and M1 macrophages induce DNA damage. (**A**) H2A.X immunostaining of undifferentiated THP1 cells after 24 h of treatment with 3MA (2.5 mM) and NAC (25 mM). Bar = 100 µm. H2A.X immunostaining (**B**) and QPCR (**C**) of M1 and M2 THP1 macrophages. During M1 polarization, cells were treated with hemin (25 μM), and during M2 polarization, cells were treated with bafilomycin (100 nM). Bar = 200 µm. (*n* = 6) (* *p*-value < 0.05, one-way ANOVA).

**Figure 5 ijms-25-00169-f005:**
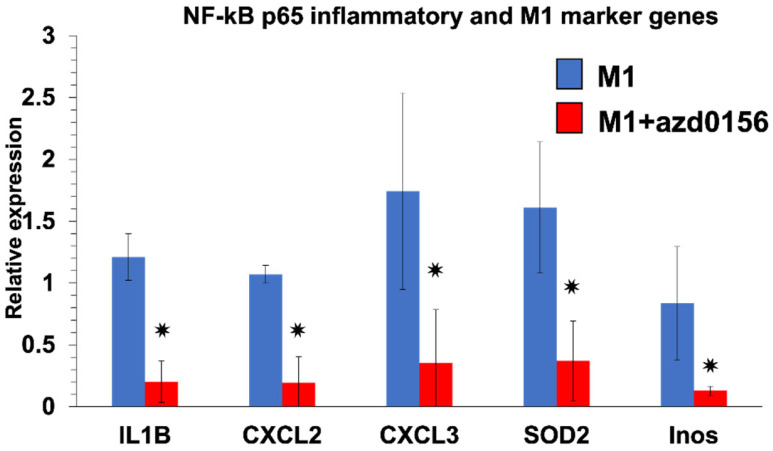
ATM inhibition alleviates the M1 phenotype and inflammation. QPCR results showing levels of expression of NF-kB p65 inflammatory and M1 marker genes in M1 THP1 cells (*n* = 3) after being treated with ATM inhibitor azd0156 during M1 polarization. All genes marked by asterisk were significantly upregulated upon azd0156 (*p*-value < 0.05, *t*-test).

**Figure 6 ijms-25-00169-f006:**
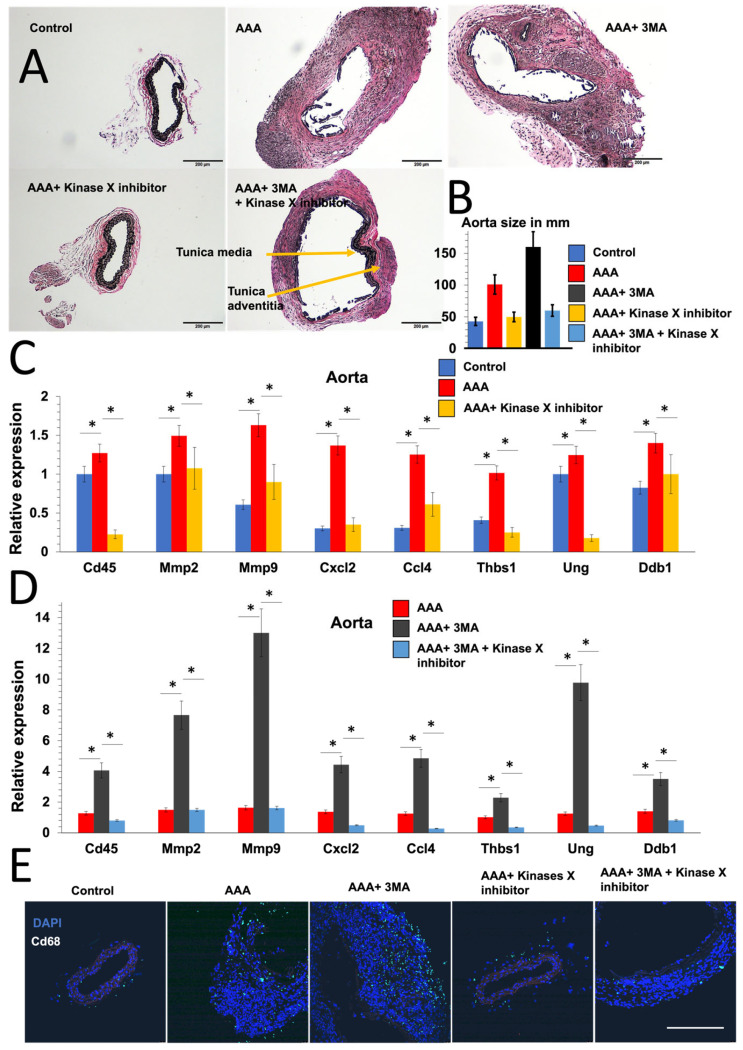
The autophagy–ATM pathway regulates AAA development in a CaCl2 AAA mouse model. (**A**) Representative images of EVG staining (Elastica van Gieson staining) of the control group, CaCl2-induced AAA mouse model, and AAA mouse model with the administration of ATM inhibitor azd0156 and/or autophagy inhibitor 3MA. (**B**) Size of aorta in mm in the five groups. (*p*-value < 0.05, *t*-test). (**C**,**D**) QPCR results showing levels of expression of leukocytic marker Cd45, matrix metalloproteinases Mmp2 and *Mmp9*, inflammatory genes Cxcl2, Ccl4, and *Thbs1*, and DNA-damage-related genes *Ung* and *Ddb1* in the five groups. Asterisks indicate a statistically significant (*p*-value < 0.05, *t*-test). (**E**) Representative images of Cd68 immunostaining of isolated aorta between the five groups. Bar = 200 μm.

## Data Availability

All relevant data generated and analyzed during this study are included in the article. The raw data supporting the conclusions of this article will be made available by the authors upon reasonable request, without undue reservation. Additional datasets used and/or analyzed during the current study are available from the corresponding author upon reasonable request.

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
