# Peer review of "Antioxidants and azd0156 Rescue Inflammatory Response in Autophagy-Impaired Macrophages"

_ijms, 2023, doi:10.3390/ijms25010169_

Round 1

Reviewer 1 Report

Comments and Suggestions for Authors

The article by Elbialy et al., sheds light on the how antioxidants rescue the effects caused by impaired autophagy specific to macrophages. The work  improves the understanding of how macrophages react to autophagy inhibition. 

The authors have looked at the levels of p62 and other markers of autophagy in presence of Baflomycin. 

However, it would have been nice if authors have used Torin1 to induce autophagy alongside Hemin. 

As the article invovles extensive use of qPCR, it would be nice if the authors include list of the primer sequences used for the study. 

Reviewer 2 Report

Comments and Suggestions for Authors

In this manuscript, the authors focus on how autophagy is linked to macrophage inflammation. Notably, autophagy is differentially influenced by M1 and M2 macrophages. Induction of autophagy in M1 macrophages regulates the expression of inflammation markers in an oxidative stress dependent manner. Inhibition of autophagy also induces DNA damage. Furthermore, the application of ATM inhibitors leads to a reversal of inflammation. Subsequently, the authors demonstrated the involvement of autophagy and ATM in the development of AAA. The paper presents an interesting angle to study the interplay between autophagy and macrophage inflammation. However, some of the conclusions are not solidly supported by the findings presented here.

Major comments

·         The first two result sections can be merged into a single section.

·         The authors claim in result section 1 that M1 macrophages suppress autophagy as evidenced by monitoring only the expression levels of p62. To solidify their findings, it would be interesting to examine additional autophagy markers such as LC3.

·         In Figure 1c, the authors demonstrate that the activation of autophagy has opposing effects on M1 and M2 macrophage differentiation, as indicated by iNOS and Arg1. To verify this, the authors should employ additional phenotypic markers of M1 (CD86) and M2 macrophages (CD206).

·         In figure 1 the authors provide the relative expression of various NF-kB p65 inflammatory genes. Please provide in supplementary Table 1 and 2 the expansions of all genes tested.

·         Authors’ interpretation heavily relies on qPCR data. In all graphs provided, the statistical significance is denoted with one * regardless of high fold changes observed in several treatments. Please confirm the accuracy of the displayed statistical differences. Furthermore, statistical significance between treatments is not presented in Figure 6.

·         Please verify that materials and methods section includes information on the in vivo experiments for AAA development.

Minor comments

·         Please correct the title legend of figure 4.

·         Please correct the syntax/typos in the text. For example: line 22 “Inhibiting”, line 36 “Identifying”, line 67  “since”.Summary

Comments on the Quality of English Language

The manuscript does not face issues related to narrative flow and language.

Reviewer 3 Report

Comments and Suggestions for Authors

Dear authors: Congratulations on this important work. 

 Monocyte/macrophage inflammation is directly involved in autoimmune diseases such as  multiple sclerosis and rheumatic arthritis.  Your results showing that M1 macrophages suppress autophagy are relevant not only in this field, but they could also be relevant in the context of viral infections. Specifically, it has been demonstrated that SARS-CoV-2 infection also supresses autophagy to increase its replication. The main protease of SARS-CoV-2, NSP5, effectively cleaves the selective autophagy receptor p62. NSP5 targets p62 for cleavage at glutamic acid 354 and thus abolishes the capacity of p62 to mediate selective autophagy. 

https://pubmed.ncbi.nlm.nih.gov/35654983/

This inhibition of autophagy induced by the virus could cause or exacerbate autoimmune conditions such as those already mentioned. In fact, there are many reports of auto-immune diseases diagnosed after SARS_COV-2 infection. 

Therefore, this excellent work provides very valuable information that can also be used to understand the complex interactions between viruses and their hosts.

Also important is the fact that you discovered that antioxidants like N-acetylcisteine (NAC) could reverse the harmful effects of autophagy suppression, and NAC has been proposed as a treatment for COVID-19.

Reviewer 4 Report

Comments and Suggestions for Authors

Description:

Elbialy E., et al., describes/work presented, how the inflammatory state of macrophages is regulated by autophagy via p65 mediated genes. Authors showed that, autophagy is impaired in M1 macrophages which leads to production of reactive oxygen species, responsible for DNA damage and causing inflammation ultimately. Further they demonstrated that use of antioxidant (e.g. NAC) or DNA damage (ATM) inhibitor (azd0156) can suppress the p65 meditated inflammatory gene expression and rescue cells from inflammatory response. The current piece of research showed interesting data and provided sufficient experimental proofs to support their hypothesis very well. This work would be physiologically relevant and applicable for some life-threatening diseases caused du to macrophages inflammation. I would advise authors to include following suggested points to make it more presentable and impactful to the readers/audience. Will be happy to consider this manuscript for publication after the minor revision. 

Major points:

1.     It is suggested to show the cytokine responses at protein level too apart from RNA level. It would be nice if author demonstrate proinflammatory cytokine levels (IL-1B, TNF-alpha or IL-6) through ELISA also.

2.     Statistics: Mention clearly which Anova (one-way or two-way) applied statistically for each experiment. It looks strange that for very comparison authors got * significance whether there is minor or major differences. 

Minor:

1.     Figure-1A: It seems the p62 immunofluorescence picture is over saturated in M1 Lower exposer of p62 required. 

2.     Figure legend -3: Please change to 3MA instead of Baf. All genes were significantly upregulated upon Bafilomycin treatment 128 (*p-value < 0.05 Anova). 

3.     Figure-6:  Please indicate the number of mice along with include statistics for panel-B, C and D. 

Round 2

Reviewer 2 Report

Comments and Suggestions for Authors

The authors have addressed my concerns. I now recommend this work for publication. 

Minor point

Please review and adjust the values displayed on the y-axis in figure 1. Ensure the axis break does not start and end with the same value. 
